# Traditional Bangladeshi Sports Video Classification Using Deep Learning Method

**Moumita Sen Sarma** [1], **Kaushik Deb** [1,*], **Pranab Kumar Dhar** [1] **and Takeshi Koshiba** [2]

1   Department of Computer Science and Engineering, Chittagong University of Engineering & Technology, Chattogram 4349, Bangladesh; moumitactg@gmail.com (M.S.S.); pranabdhar81@cuet.ac.bd (P.K.D.)
2   Faculty of Education and Integrated Arts and Sciences, Waseda University, 1-6-1 Nishiwaseda, Shinjuku-ku, Tokyo 169-8050, Japan; tkoshiba@waseda.jp
*   Correspondence: debkaushik99@cuet.ac.bd

**Abstract:** Sports activities play a crucial role in preserving our health and mind. Due to the rapid growth of sports video repositories, automatized classification has become essential for easy access and retrieval, content-based recommendations, contextual advertising, etc. Traditional Bangladeshi sport is a genre of sports that bears the cultural significance of Bangladesh. Classification of this genre can act as a catalyst in reviving their lost dignity. In this paper, the Deep Learning method is utilized to classify traditional Bangladeshi sports videos by extracting both the spatial and temporal features from the videos. In this regard, a new Traditional Bangladeshi Sports Video (TBSV) dataset is constructed containing five classes: Boli Khela, Kabaddi, Lathi Khela, Kho Kho, and Nouka Baich. A key contribution of this paper is to develop a scratch model by incorporating the two most prominent deep learning algorithms: convolutional neural network (CNN) and long short term memory (LSTM). Moreover, the transfer learning approach with the fine-tuned VGG19 and LSTM is used for TBSV classification. Furthermore, the proposed model is assessed over four challenging datasets: KTH, UCF-11, UCF-101, and UCF Sports. This model outperforms some recent works on these datasets while showing 99% average accuracy on the TBSV dataset.

**Keywords:** traditional Bangladeshi sports; convolutional neural network; long short term memory; transfer learning; fine tuning

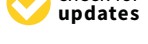

## 1. Introduction

Sports activities have an immense impact on a person's mind, body, and spirit. They not only alleviate stress and anxiety, but also fill our soul with positive energy. Accelerated by the tremendous technological growth, sports video data was produced, distributed, and explosively dispersed, becoming an invaluable part of big data of this age. This stream has driven the evolution of upgraded techniques for a wide range of sports video related applications that require a perception of those video contents. Classifying sports videos into different genres or groups is an essential activity, allowing effective indexing and retrieval from extensive video collections. This task can be widely used in recommendation systems that suggest sports videos to the end-users and also in promoting sports commercials in streaming services as per the user's interests. The objective of sports video classification focuses on the automated tagging of sports video clips based on apposite and specific sports action, consisting of spatial and motion features. Computer vision and artificial intelligence technologies are becoming trendy in the study of analyzing videos. However, the domain of computer vision is currently shifting from statistical approaches to deep learning methods due to their efficiency in extracting complex features without human intervention. DNN or deep neural network-based models have been widely used and effectively solve intricate signal processing, machine translation, and computer vision tasks. Therefore, there is a huge scope for commencing innovation and development in this evolving research arena. Furthermore, the two most prominent deep learning networks,

i.e., convolutional neural network (CNN) and recurrent neural network (RNN), can capture the spatial and temporal features, respectively, from videos that are obligatory for correctly classifying the video classes. Additionally, sports video classification is quite a novel research interest, and there is still room for improvement and exploration on this ground.

The history of Bangladesh encompasses its cultural diversity and literary heritage. Furthermore, significant parts of its profoundly ingrained heritage are reflected in its traditional sports activities. Therefore, sport in Bangladesh is a popular medium of amusement as well as an integral part of Bangladeshi culture. However, due to the lack of nurture and retention, these traditional sports activities are losing their dignity day by day. Classification of traditional Bangladeshi sports videos perhaps can revive the glory and pride of these sports.

In this paper, our main concern is to classify five traditional Bangladeshi sports categories, i.e., Boli Khela, Kabaddi, Lathi Khela, Kho Kho, and Nouka Baich, from a newly constructed dataset of 500 traditional Bangladeshi sports videos through incorporating the convolutional neural network with the recurrent neural network. A variation of RNN, namely long short term memory (LSTM), is explored and integrated with the CNN architecture. Additionally, we have experimented with a widely used deep learning technique, the transfer learning method, by exploring pretrained VGG16 and VGG19 networks combined with RNN to accomplish the traditional Bangladeshi sports video classification task. Through extensive exploration and evaluation, the optimum model, i.e., the proposed model, has been attained by integrating the fine-tuned VGG19 network with the LSTM layer. In addition, we have extensively assessed our proposed model in the four most challenging and eminent datasets, KTH Sports, UCF-11, UCF Sports, and UCF-101. The proposed model exhibits impressive performance in these datasets relative to some existing approaches. However, the key contributions of this paper are enlisted as follows:

- Developing a novel Traditional Bangladeshi Sports Video (TBSV) dataset containing 500 traditional Bangladeshi sports videos belonging to five classes: Kabaddi, Boli Khela, Kho Kho, Lathi Khela, and Nouka Baich.
- Building a scratch model combining CNN and LSTM to classify the five sports classes from the TBSV dataset.
- Exploring the transfer learning approach in this task through training some popular and recent pretrained models: ResNet, Inception v3, Xception, VGG, incorporating with LSTM, and comparing their performance.
- Fine tuning the two best performing pretrained models, VGG16 and VGG19, integrated with some custom layers of LSTM and dense layer.
- Assessing the proposed model, fine-tuned last eight layers of VGG19 incorporated with LSTM, over four challenging datasets, KTH, UCF-11, UCF-101, and UCF Sports, and comparing performance with related recent works.

## 2. Related Work

As previously stated, sports video classification is an entirely new arena in the emerging field of deep learning technologies. This concept has drawn the attention of some researchers who were motivated by the novelty of this task. On the other hand, before the evolution of deep learning techniques, various statistical and classical methods were used by some researchers for categorizing sports videos. However, a few existing methodologies are found for classifying Traditional Bangladeshi Sports Video, as the extent of this task is still dormant.

In the early stage, various statistical and classical methods were used to extract the features from inputs manually. The extracted features lead to the final classification of different categories of videos. Cricri et al. [1] analyzed sensor, audio, and visual data to classify sports categories from user-generated mobile videos and explored multi-user and multimodal data along with a multiclass SVM classifier. However, sports having similar semantic views encountered some misclassifications. Edge features of 500 sports videos obtained from Non-Subsampled Shearlet Transform (NSST) were analyzed by Uma Mah-

eswari and Ramakrishnan [2], and KNN was used as a classifier to categorize five sports video types. Additionally, under the ASSAVID system, Messer et al. [3] proposed an automatic sports image classification method through KNN classifier, yet some misclassification was noted.

On the other hand, Gade and Moeslund [4] categorized sports video types from signature heatmaps by Fisherfaces algorithm as PCA, and Euclidean distance as means of classifier focusing only on the sport played in match-like situations. Gade et al. [5] considered audio and visual features captured from the thermal video to classify 180 video sequences of 1 min containing three sports types. Here, four motion features were extracted and integrated with MFCC audio features, and KNN was used as the classifier. However, Gibert et al. [6] fused two HMMs representing color and motion features and used the Baum–Welch algorithm to classify 220 minutes of sports video with four genre types. Likewise, Hanna et al. [7] proposed a Hidden Markov Model (HMM)-based classification technique integrated with Baum–Welch algorithm to categorize three genres of sports videos where the speed of color changes was computed for each video frame. Wang et al. [8] explored dominant color, global camera motion, and MFCC features for audio-visual analysis of three categories of sports videos. They used the HMM-based model for classifying them. Ellappan and Rajasekaran [9] classified four different events, i.e., goal-kick, placed-kick, shot-on-goal, and throw-in of soccer video dataset through observing ball directions using HMM. Moreover, the time performance of their experiment on the soccer event classification module was also represented.

In recent times, Deep Learning methods have been proven to extract complex features from images automatically. In this context, several researchers have worked on the two most recent deep learning techniques, convolutional neural network (CNN) and recurrent neural network (RNN), to extract spatial and temporal features from sports videos. Russo et al. [10] introduced a combined architecture of dilated CNN and RNN for classifying five categories of sports from sequential frames, and the performance of the architecture was analyzed according to different frame sequences. However, more high-level features could not be extracted due to the inability to implement deep architecture with large enough input size because of hardware limitations. Moreover, in the extension of their research work, Russo et al. [11] proposed two types of scratch model, combining CNN and RNN and a transfer learning approach with pre-trained VGG16 model along with RNN on two sets of comprehensive data named SportsC10 and SportsC15, containing 10 and 15 categories of sports, respectively. In this case, the transfer learning approach integrated with RNN came up with the most promising performance.

Carrying out a comparative analysis of several feature extraction methods and classifiers, Campr et al. [12] used CNN and median filter with various window lengths to filter classification results for frame sequences in continuous TV broadcasts. Though their system can be used both in real-time and offline applications, better performance was achieved offline, and misclassification among visually and semantically similar sports was also observed. Furthermore, extensive research was conducted by Karpathy et al. [13], which prolonged the connectivity of CNN in the time domain with noteworthy performance improvement relative to strong feature-based baselines. They introduced a multiresolution, foveated architecture for speeding up the training, and three types of fusion models were implemented on a dataset of 1 M YouTube videos having 487 sports classes. However, Ye et al. [14] constructed a two-stream convolutional neural network (CNN) pipeline by using both static frames and motion optical flows and implemented them on two baseline datasets, i.e., CCV and UCF-101. Jiang et al. [15] established a deep neural network combining RNN and CNN to detect soccer video events and built a dataset named Soccer Semantic Image Dataset (SSID). Finally, RNN is used to map the extracted semantic features to four categories of soccer events, i.e., card, corner, goal, and goal attempt.

Likewise, Ullah et al. [16] established an action recognition framework on three prominent datasets: UCF-101, YouTube 11 Actions, and HMDB51, through extracting in-depth features from every sixth frame of the videos by utilizing CNN and Bi-directional

LSTM, which yielded improved results over state-of-the-art action recognition methods. On the other hand, the AlexNet architecture was used by Minhas et al. [17] for four types of shot classification categorized as close-up, long, medium, and crowd/out-of-the-field shots in field sports videos of cricket and soccer collected from YouTube. Additionally, Jaouedi et al. [18] constructed a hybrid model by combining GMM and Kalman Filters with GRU in classification tasks over KTH, UCF 101, and UCF Sports datasets. This model produced promising results relative to some existing approaches on these datasets. By analyzing the temporal coherence, Ge et al. [19] proposed an attention mechanism based GoogleNet-LSTM model that showed impressive performance to recognize actions in UCF-11, HMDB-51, and UCF-101 datasets. Their method was capable of effectively grabbing the salient areas of actions in videos. However, Abdelbaky and Aly [20] used spatio-temporal features learned from CNN based principal component analysis network (PCANet) with bag-of-features and encoding schemes in KTH and UCF Sports action recognition dataset and attained promising results. They calculated temporal templates through short-time motion energy images (ST-MEI) as per frame differencing. Likewise, Yang et al. [21] introduced a novel recurrent attention CNN for focusing the crucial regions in videos to detect actions in UCF-11, HMDB-51, and UCF-101 datasets. Their "attention-again" model incorporated the adjacent frames with current one for obtaining the crucial part of the frame.

On the other hand, considering the overfitting problem of deep neural networks, Martin et al. [22] proposed an ensemble model named gaNet-C for IoT type-of-traffic forecast problem to detect forecast of active connections and elephant flows of networks. They implemented some variants of gaNet-C, and the proposed two variants hold a built-in regularization feature that reduced the overfitting problem to a great extent.

However, a very minimal amount of significant research has been carried out in recognizing traditional Bengali sports. Islam et al. [23] used the transfer learning approach through retraining the last layer of eminent Inception V3 model by Google for classifying five traditional Bengali sports categories, i.e., "Danguli", "Kabadi", "Kanamachi", "Latthi Khela", and "Nouka Baich", from augmented images of a dataset having in total 3600 images, yielding decent results in this ground.

Therefore, it can be said that the classification of sports categories has been bearing great significance for years. Moreover, in recent times, deep learning models integrated with a wide variety of configurations have outperformed existing state-of-the-art approaches. Therefore, there is a huge scope for commencing innovation and development in this evolving research arena. Furthermore, implementation of this task on the Traditional Bangladeshi Sports Video dataset is quite a fresh conception, which seemed quite fascinating to work with, and it will be an endeavor to revive the traditional extinct Bangladeshi sports.

## 3. Proposed Approach

In the video, there exists a correlation between the subsequent frames in terms of their semantic contents. Therefore, if the temporal connections of the frames can be tracked, the classification system will be able to attain a promising result. Hence, it is a prerequisite to pick a specific number of subsequent frames for the analysis of the video data. In this context, at first, a particular number of sequential RGB color frames are selected from the input sports video. Next, some preprocessing operations, resizing, rescaling, augmentation, and normalization, are performed to make the raw frames suitable for further processing. Then, spatial features are captured simultaneously from the processed frames of a video through CNN. After that, the extracted features by CNN are passed to the LSTM layer to capture the temporal features. Finally, a Softmax activation function-based output layer preceded by a fully connected layer is used to classify the sports class. Figure 1 shows the steps of the proposed approach for traditional Bangladeshi sports video classification workflow.

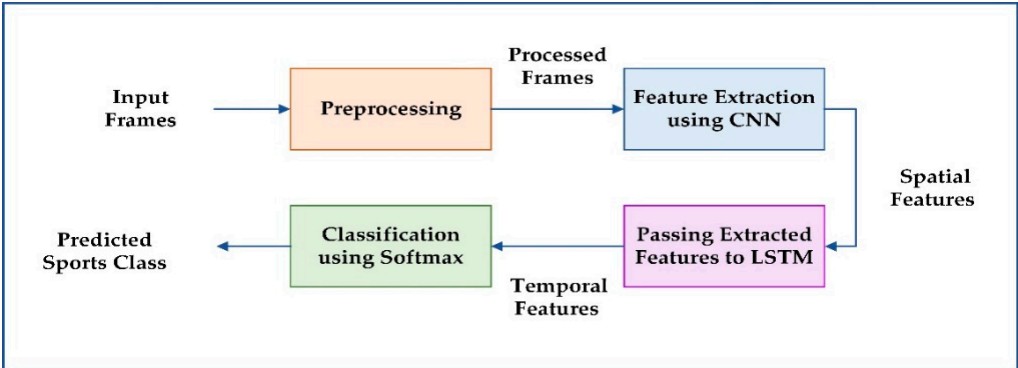

**Figure 1.** Workflow of the proposed TBSV classification approach.

### 3.1. Preprocessing

Image processing refers to improving image data (properties) by removing unnecessary artifacts and enhancing certain essential image features so that the deep learning methods can take advantage of this improved data. In this paper, preprocessing is the first step of the Traditional Bangladeshi Sports Video Classification framework. However, the preprocessing step consists of three segments: resizing and scaling, augmentation, and normalization. Images/frames are resized to $128 \times 128$ pixels to reduce computational cost and time, which leads to the boosted performance of the model. Moreover, the original images consist of RGB coefficients in the 0–255 range that will be too high to be processed. Therefore, we have converted the pixel values to the [0, 1] interval by scaling with a $1/255.0$ factor.

On the other hand, augmentation refers to generating more training instances from the prevailing ones through some specified transformations for raising up the generalization power of the deep learning models. In this work, online, i.e., real-time augmentation, has been used on resized and rescaled training images, which creates transformed images at each epoch of training. In this regard, we have utilized four different transformation methods: horizontal flip, width shift, height shift, and rotation. This mechanism leads to a better generalization of the model.

Normalization is added to standardize raw input pixels, which will transform input pixels by making mean 0 and standard deviation of 1. If data are not normalized, there may be some numerical data points in our data set that might be very high and others that might be very low. In unnormalized data, thus naturally, large values become dominant according to relatively small values during training, as the importance of each input is not equally distributed. When we have normalized our training data, however, we have put all of our data on the same scale, which makes convergence faster while training the network. Effect of normalization on the performance of the model is briefly explained in this paper.

### 3.2. Spatial Feature Extraction

In videos, spatial features or elements can be defined as the characteristics relevant to the context of the scenes. However, in the case of sports videos, spatial features include surrounding aspects of the sports ground, characteristics of the athletes, and all about the event view. In recent times, CNN has proven its prosperity to a great extent in extracting complex spatial features of the images. The integral component of the CNN is the convolutional layer that takes after the name of this network. Convolutional layers can learn local and translation invariant patterns from the images through "convolution". It denotes a linear mathematical operation that performs matrix multiplication between the filter of a specific dimension and the portion of the image on which the filter hovers. The outcome of a convolutional layer is termed as "feature map", passed to an activation function that introduces nonlinearity in the output. Another integral element of CNN is the pooling layer that has come up with a downsampling strategy. The key purpose of this layer is to accumulate the most activated presence of a feature by gradually decreasing the spatial

size of the representation. However, for extracting the spatial features from traditional Bangladeshi sports videos, a TimeDistributed CNN architecture has been constructed. The input dimension of a TimeDistributed CNN layer is (samples, frames per video, height, width, channels). This architecture employs the same layer of this architecture to all the picked frames of each video one by one through sharing same weights and thus finally produces feature maps for the frames.

### 3.3. Temporal Feature Extraction

There remains temporal connectivity across sequential frames in sports videos. The recurrent neural network may play a crucial role in this regard. In RNN, not just the present input but also previously obtained inputs are taken into account to record the motion shifts in the successive frames. However, traditional RNN suffers from short term memory, i.e., it is incompetent in retaining information for longer periods. In this regard, LSTM, i.e., long short term memory, was developed as a remedy to short-term memory and vanishing gradient problems. LSTM was introduced to capture long term dependencies in sequential data with more controlling power on information flow. Moreover, LSTM resolves the vanishing gradient problem, as it possesses an additive gradient mechanism, which is contrary to the multiplicative gradient process of basic RNN that diminishes over the long sequence. However, it consists of a cell state and three gates, i.e., input gate, forget gate, and output gate. The cell state retains salient information over a time interval, and the gates control the information flow across the cell. The input gate determines which new information is going to be saved by the cell state. On the other hand, the forget gate elects which information will be eliminated from the cell state. Additionally, the output gate determines the outcome of the current LSTM cell, named as the hidden state. The equations used for the input gate, forget gate, cell state, and output gate are given below:

$$i_t = \sigma \left( W_i \cdot cnt \left( H_{t-1}, \, X_t \right) + b_i \right) \tag{1}$$

$$f_t = \sigma \left( W_f \cdot cnt \left( H_{t-1}, \, X_t \right) + b_f \right) \tag{2}$$

$$C_t = f_t * C_{t-1} + i_{t-1} * tanh \left( W_c \cdot cnt \left( H_{t-1}, \, X_t \right) + b_c \right) \tag{3}$$

$$o_t = \sigma \left( W_o \cdot cnt \left( H_{t-1}, \, X_t \right) + b_o \right) * tanh \left( C_t \right) \tag{4}$$

In this paper, the extracted spatial features of sequential frames by CNN are fed to an LSTM layer for analyzing them in order.

In the above equations, $i_t$, $f_t$, $C_t$, and $o_t$ refer to input gate, forget gate, cell state, and output gate, respectively. $W_i$, $W_f$, $W_c$, and $W_o$ are weight matrices, and $b_i$, $b_f$, and $b_c$ are bias values of input gate, forget gate, cell state, and output gate, respectively. Additionally, $H_{t-1}$ refers to the previous hidden state, $X_t$ is the current input, $*$ signifies pointwise multiplication, *cnt* stands for the concatenation operation, and $\sigma$ and *tanh* represent activation functions.

### 3.4. Classification Using Softmax

For classification, the Softmax activation function is used in the output layer that produces the probability distribution of the five classes based on the extracted features of the previous step. This function outputs probabilistic values ranging between 0 to 1, all summing up to 1. Softmax activation function generates probability distribution by using the following equation:

$$S(\vec{i})_k = \frac{e^{i_k}}{\sum_{n=1}^{t} e^{i_n}} \tag{5}$$

In this equation, $\vec{i}$ denotes the input vector, $k$ is the index of the current element in the input vector, all the $i$ values refer to the elements of the input vector, and $t$ represents the total number of classes.

### 3.5. Scratch Model

Building a deep neural network from scratch is essential for a better understanding of the mechanism of deep learning methods and getting insight into the dataset's feature space. In this paper, a scratch model is developed by incorporating CNN with LSTM. After applying the preprocessing tasks, the processed frames of video are passed to the CNN architecture, used as a spatial feature extractor. The CNN architecture is comprised of nine layers containing convolution layers with a varied number of filters of $3 \times 3$ dimension, maxpool of $2 \times 2$ pool size with stride 2, and flatten layers. In this architecture, the "same-padding" is used, and the stride dimension is (1, 1). For faster convergence during training, ReLU is used as an activation function that provides output to the $[0, \infty]$ interval. Afterwards, a flatten layer is appended to create a single long feature vector for each selected frame of a video. This CNN architecture processes the sequentially picked frames of a video, and features are extracted from those frames. In this regard, we have implemented the CNN part our model using TimeDistributed layer of Keras. This wrapper employs a same layer to all the selected sequential frames of a video one by one and produces feature vector for each frame. The layer, which is TimeDistributed, shares same weights while processing the frames of a video. Without a TimeDistributed wrapper, a layer updates weights while working on each frame of a video, which resembles image classification. This method lacks the consideration of frames as sequential parts of a video and thus is unable to grab the features of the frames with respect to the whole video. Later, an LSTM layer of 128 hidden units is utilized to extract the temporal features from the feature vector produced by CNN architecture. A dense layer of 128 neurons is added next with a dropout rate of 0.2, followed by the output layer, where the Softmax activation function is used to classify five individual traditional Bangladeshi sports video classes. Therefore, in total, the scratch model contains 17,248,843 weights/parameters. Scratch model architecture is depicted in Figure 2.

### 3.6. Transfer Learning Approach

As deep neural networks demand a vast amount of training data for learning the features, it is quite an effective approach to use a pretrained model in the case of working with a relatively small dataset. This concept is termed as Transfer Learning approach. Transfer learning can extensively reduce the training time required for the deep learning model and produce low generalization error. Moreover, it is a versatile technique that invokes the use of pre-trained models as feature extraction methods that can be incorporated into entirely new models. There are several numbers of top-notch pretrained models that were constructed for computer vision tasks. Inception v3, VGG, Xception, and ResNet are such pretrained models, trained on one of the benchmark datasets, ImageNet, and rendered impressive performance on the ImageNet Classification Challenge. Additionally, VGG16 and VGG19 are two versions of the VGG network containing about 138 and 143 M parameters, respectively. However, a widely utilized tactic for model reuse is fine-tuning that aims to make the pretrained model more relevant for the new dataset. Fine-tuning can be performed through unfreezing some of the top layers of a frozen model, adding some custom layers over that model, and then jointly training both the unfreezed and custom layers. In this paper, we have explored pretrained Inception v3, VGG16, VGG19, Xception, and ResNet network on the TBSV dataset for extracting spatial features integrated with LSTM. Inspired by the better performance of VGG16 and VGG19, we dived into further exploration of these networks. In this regard, we have conducted an experiment with the fine-tuning approach of VGG16 and VGG19, incorporated with LSTM. Performance comparison of the inspected pretrained models and various configurations of fine-tuned VGG16 and VGG19 are briefly illustrated in this paper.

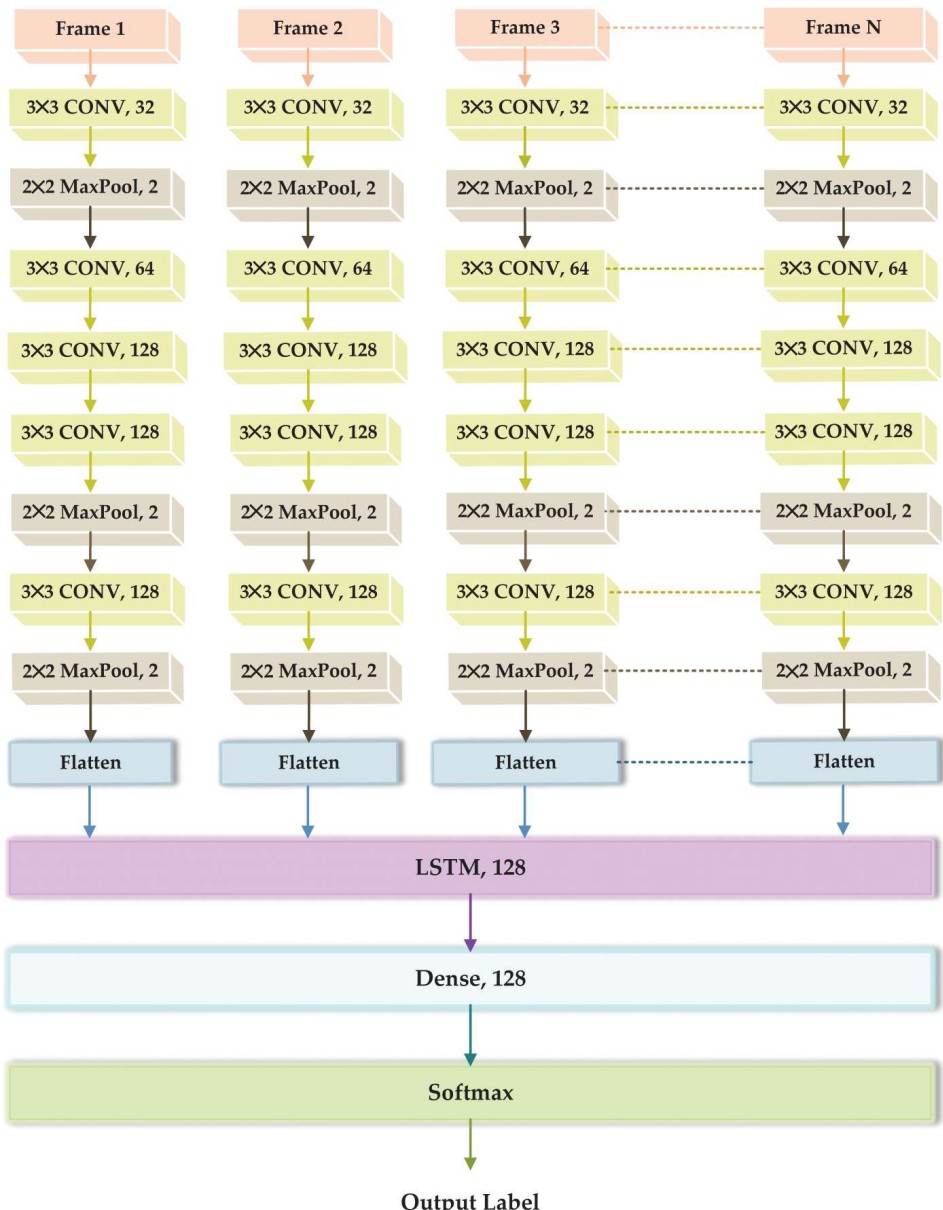

**Figure 2.** The architecture of the scratch model.

## 4. Experiments

The whole experiment was conducted in a multicore processor equipped with Tesla K80 GPU (NVIDIA, Santa Clara, CA, USA) with a 25 GB memory and 2.3 GHz processor. As the deep learning framework, Keras, along with Tensorflow v2.0, was used with Python v3.6.

### 4.1. Dataset Description

We conducted our experiment over four datasets, our newly self-developed TBSV dataset, KTH [24], UCF-11 [25], UCF-101 [26], and UCF Sports [27,28] dataset, to justify the integrity of our proposed model.

#### 4.1.1. TBSV Dataset

As mentioned earlier, a few attempts can be found to classify traditional Bangladeshi sports videos. However, unfortunately, there is no standard dataset for this task. Therefore, one of the vital contributions of this work is to develop the novel TBSV dataset. This dataset

consists of 500 traditional Bangladeshi sports videos belonging to five classes: Kabaddi, Boli Khela, Kho Kho, Lathi Khela, and Nouka Baich, collected from Youtube. The details of this dataset are illustrated in Table 1. Examples of sequential frames of five classes are portrayed in Figure 3.

**Table 1.** TBSV dataset details.

| Traditional Bangladeshi Sports Video (TBSV) Dataset | |
| --- | --- |
| Total Classes | 5 |
| Number of videos per class | 100 |
| Frames per second | 30 |
| Video Length | 5 s |
| Resolution | 720 × 1280 |

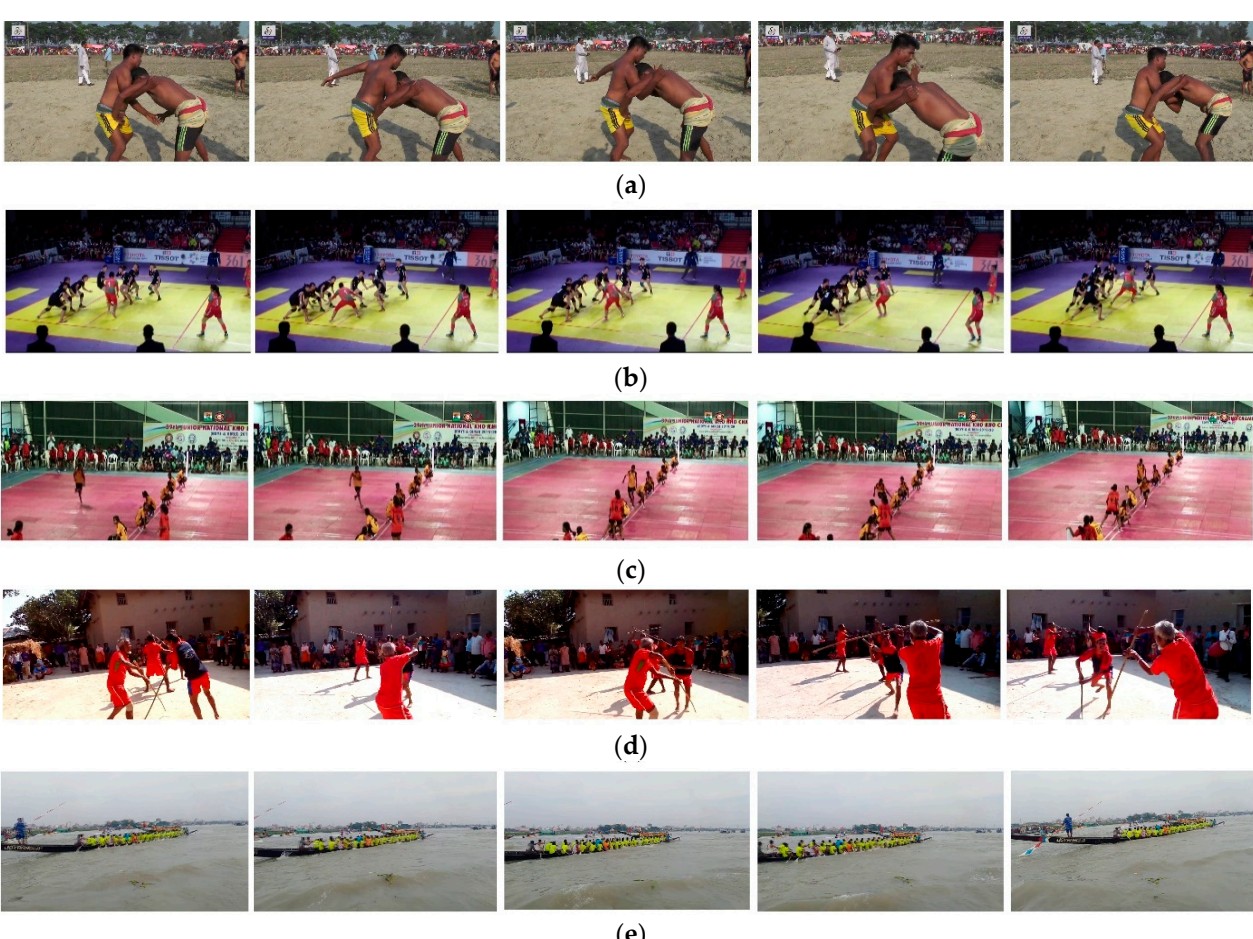

**Figure 3.** Sequential frames of (**a**) Boli Khela; (**b**) Kabaddi; (**c**) Kho Kho; (**d**) Lathi Khela; (**e**) Nouka Baich.

### 4.1.2. KTH Dataset

KTH is among the most extensive datasets, widely used in human action recognition tasks. It includes six types of human actions: walking, running, boxing, jogging, handwaving, and hand-clapping. This dataset contains 600 videos, 100 videos per class, with a resolution of 160 × 120. The videos are an average of 4 s in length and have a frame rate of 25 fps.

### 4.1.3. UCF-11

UCF-11 is one of the most remarkable benchmark datasets of action recognition. It's also named as UCF YouTube Action dataset. This dataset includes 11 action classes: biking,

basketball shooting, volleyball spiking, diving, horseback riding, soccer juggling, swinging, golf swinging, tennis swinging, trampoline jumping, and walking with a dog. It contains 1600 videos with the frame rate of 29.97 fps.

### 4.1.4. UCF-101

UCF-101 is the greatest and one of the benchmark datasets that includes 101 action classes of five categories: sports, body-motion only, human–object interaction, playing musical instruments, and human–human interaction. It contains 13,320 videos of $320 \times 240$ resolution, and the maximum number of frames is 150 per video. In this paper, due to GPU memory constraint, we have conducted our experiment over 66 classes belonging to two categories of this dataset: body-motion only and sports.

### 4.1.5. UCF Sports

UCF Sports is one of the leading datasets in the application of action recognition and localization tasks. It includes 150 sequences of 10 classes: walking, running, lifting, diving, kicking, golf-swing, swing-bench, riding-horse, swing-side, and skate-boarding, collected from YouTube. The resolution of the videos is $720 \times 480$, with a variable number of videos in each class. The frame rate of videos is 10 fps. However, the length of the videos is not fixed, whereas the min video length is 2.20 s, and the max video length is 14.4 s.

### 4.2. Evaluation Metrics

In order to assess the integrity of the mathematical or machine learning models, evaluation metrics are widely used. In the paper, to test the scratch and fine-tuned pretrained models, the five most significant evaluation metrics, accuracy, specificity, precision, sensitivity, and f1 score, were used. These metrics are defined as follows:

$$A_c = \frac{TP + TN}{TP + FP + TN + FN} \tag{6}$$

$$S_p = \frac{TN}{TN + FP} \tag{7}$$

$$P_r = \frac{TP}{TP + FP} \tag{8}$$

$$S_t = \frac{TP}{TP + FN} \tag{9}$$

$$F1\ Score = 2 \times \frac{P_r \times S_t}{P_r + S_t} \tag{10}$$

In the above equations, $A_c$, $S_p$, $P_r$, and $S_t$ denote accuracy, specificity, precision, and sensitivity. Additionally, $TP$, $FP$, $TN$, and $FN$ refer to true positive, false positive, true negative, and false negative.

### 4.3. Results and Discussion

For implementing the scratch model and the fine-tuned pretrained models in classifying the sports videos, the TBSV dataset was split into the train, test, and validation data with the ratio of 70:15:15. That means 70%, 15%, and 15% of the videos per class were randomly assigned for training, testing, and validating the models. Moreover, a specific number of frames is needed to be picked from each video to commence the task of classification. In this regard, we carried out an experiment using the scratch model to determine the right frame length, with 100 video samples in each class. In Figure 4, validation and test accuracy for frame length 10–25 are shown. From this figure, it can be observed that the model exhibits the best result, i.e., near about 93% validation accuracy and 91% test accuracy for the case of frame length 20. Hence, the frame length of 20 was deemed for further analysis and exploration in this ground.

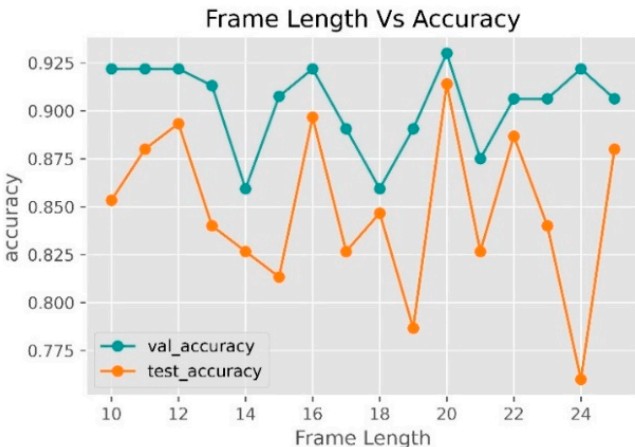

**Figure 4.** Frame length vs. accuracy curve.

The dimension of the frames is resized to 128 × 128 × 3 for the sake of reducing computational complexity. The augmentation techniques horizontal flipping, rotation, width shifting, and height shifting are applied for better generalization and avoiding positional bias in the frames. Examples of resized and augmented frames are shown in Figure 5.

As stated previously, we have used normalization for faster convergence of the model during training. The performance comparison of the scratch model on normalized and unnormalized data are represented in Table 2. The table shows that, with normalized training data, better performance has been achieved in less epoch relative to the unnormalized training data.

**Table 2.** Effect of normalization on performance.

|  | Training Accuracy | Validation Accuracy | Test Accuracy | Total Epochs |
| --- | --- | --- | --- | --- |
| Normalize Data | 99.30% | 93% | 91% | 65 |
| Unnormalize Data | 96% | 90% | 88% | 70 |

After rigorous inspection and exploration with various combinations of hyperparameters, we have come up with our scratch model that showed a splendid performance in classifying the traditional Bangladeshi sports classes. This scratch model was formed by combining CNN with LSTM, compiled through Adam optimizer with 0.0001 learning rate and categorical cross-entropy loss function. Categorical cross-entropy is a type of loss function where the target values for multiclass classification are represented in a one-hot vector. The multinomial probability distribution provided by the Softmax activation function in the output layer is used by the categorical cross-entropy loss function for measuring the prediction error/loss of the model. This function computes the loss in the following manner:

$$CCL = - log\ p(k) \tag{11}$$

In the above equation, $CCL$ refers to the categorical cross-entropy loss, and $p(k)$ denotes the probabilistic value of the class $k$ that is fired-up in the one-hot vector. However, the model is trained till 65 epochs with batch size of 64. The training and validation accuracy, as well as the loss curves of this scratch model, is depicted in Figure 6. This figure illustrates that after the 54th epoch, the training accuracy stops increasing, whereas, at this epoch, we got the maximum validation accuracy, i.e., 93%. This observation refers to the fact that at the 54th epoch, the model performs the best, and after this epoch, the model ceases learning. Moreover, Figure 7 represents the confusion matrix of this model applied to test data. It can be observed that a tiny amount of misclassification has been found in

some classes due to having some semantic feature similarity with the misclassified classes. Class-wise performance based on different evaluation metrics applied to test data for this model is summarized in Table 3. However, the overall performance is assessed through the F1 score throughout the experiments.

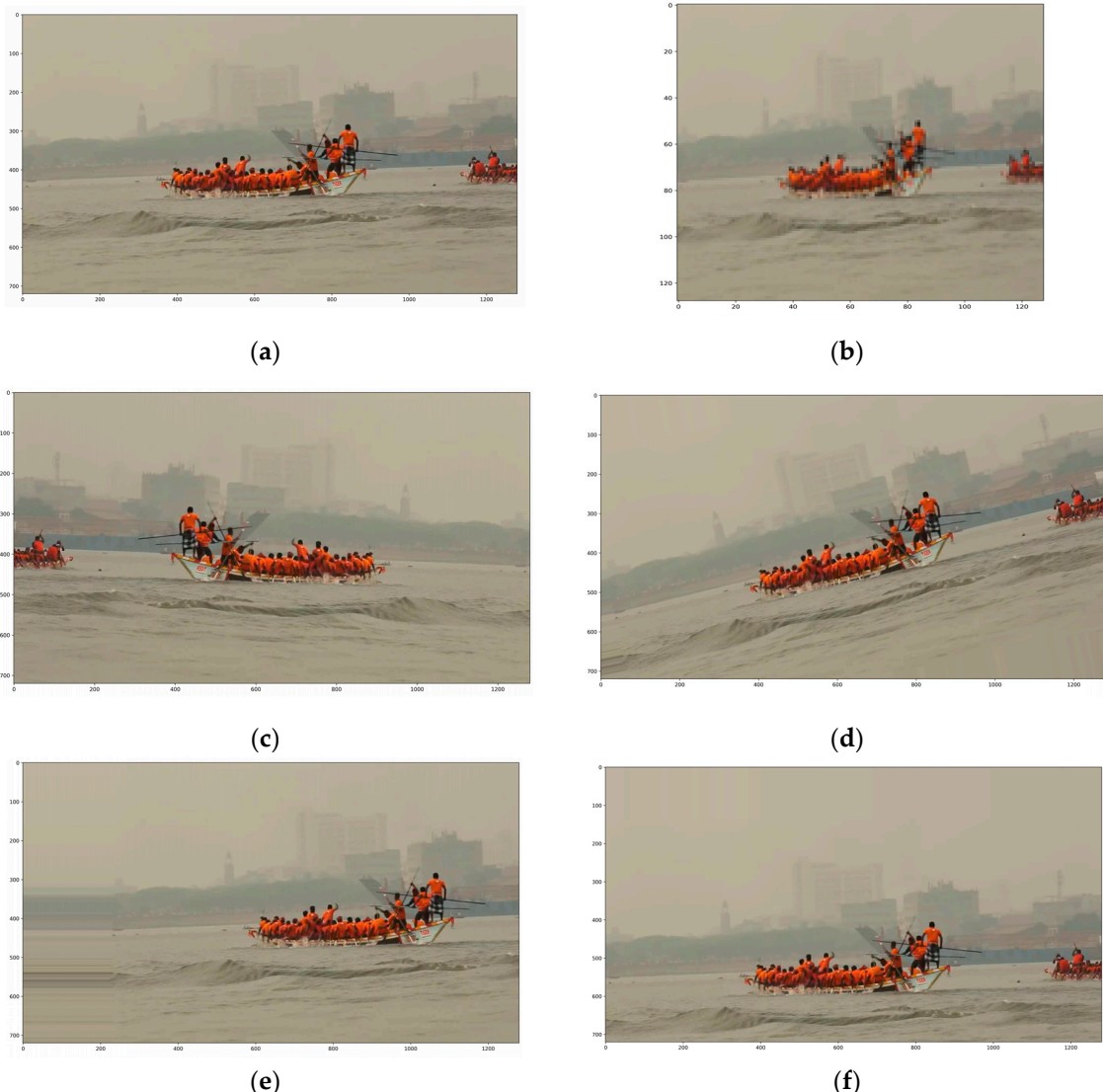

**Figure 5.** Examples of resized and augmented frames: (**a**) original frame, (**b**) resized frame, (**c**) horizontally flipped frame, (**d**) rotated frame, (**e**) width shifted frame, (**f**) and height shifted frame.

**Table 3.** Class-wise performance of the scratch model.

| Classes | Mean Accuracy (%) | Precision (%) | Sensitivity (%) | Specificity (%) | F1 Score (%) |
|---|---|---|---|---|---|
| Boli Khela | | 93 | 87 | 98 | 90 |
| Kabaddi | | 94 | 100 | 98 | 97 |
| Kho Kho | 91 | 93 | 93 | 98 | 93 |
| Lathi Khela | | 76 | 87 | 93 | 81 |
| Nouka Baich | | 100 | 87 | 100 | 93 |

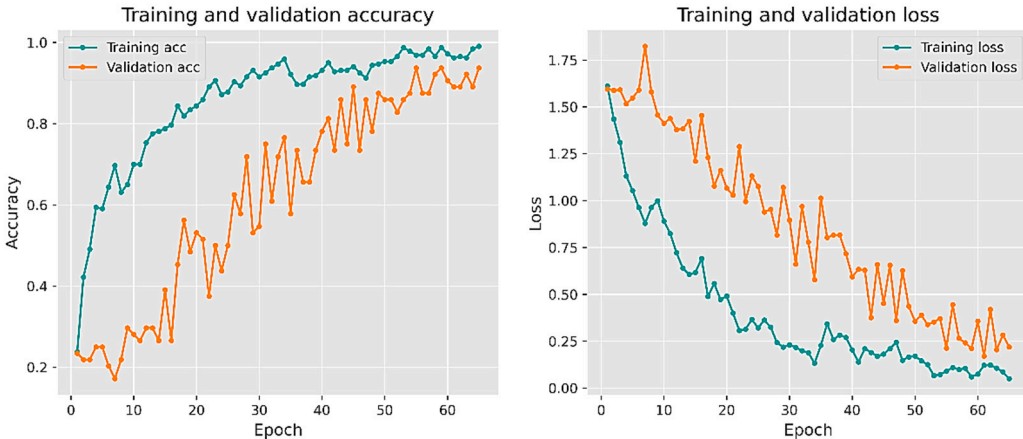

**Figure 6.** Performance of scratch model.

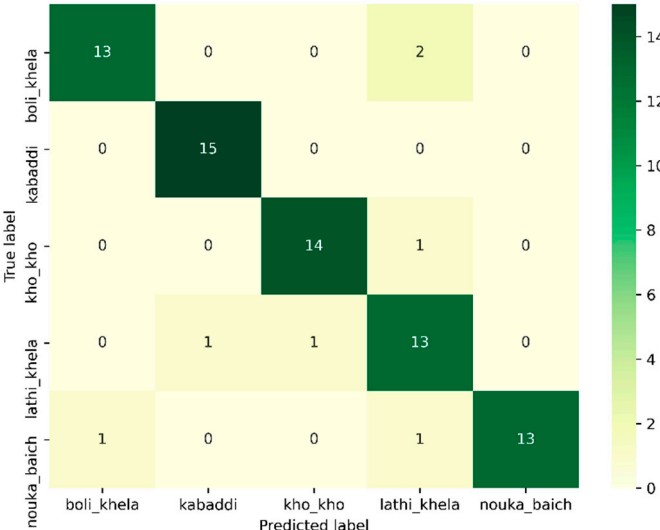

**Figure 7.** Confusion matrix for the scratch model.

Moreover, we conducted an experiment with some of the leading pretrained models combined with LSTM. The performance comparison of these models over the TBSV dataset is rendered in Table 4. From this table, it can be observed that the VGG16 and VGG19 model outperforms the others in classifying TBSV classes.

**Table 4.** Performance comparison of some pretrained models on the TBSV dataset.

| Pretrained Models | Hidden Units in LSTM Layer | Neurons in Dense Layer (0.2 Dropout) | Test Accuracy (%) |
|---|---|---|---|
| ResNet152 v2 | | | 85 |
| Inception v3 | | | 86 |
| Xception | 128 | 128 | 88 |
| VGG16 | | | 90 |
| VGG19 | | | 92 |

Furthermore, we have also explored fine-tuned VGG16 and VGG19 networks incorporated with LSTM. Various network configurations experimented with by the fine-tuning task are illustrated in Table 5. All the models are compiled using Adam optimizer with 0.0001 learning rate and categorical cross-entropy loss function, trained until 30 epochs with the batch size of 64. However, GPU memory constraint is a crucial issue in training deep learning models.

**Table 5.** Network configuration and f1 score of fine-tuned VGG models.

| Model | VGG Network Version | Trainable Layers of VGG (Excluding fc Layers) | Hidden Units in LSTM Layer | Neurons in Dense Layer (0.2 Dropout) | Total Trainable Parameters | Training Accuracy (%) | Average F1 Score (%) |
|---|---|---|---|---|---|---|---|
| Model 1 | VGG-16 | None | | | 4,277,515 | 99 | 90 |
| Model 2 | VGG-16 | All | | | 18,992,203 | 97 | 91 |
| Model 3 | VGG-16 | Last 4 layers | | | 11,356,939 | 100 | 97 |
| Model 4 | VGG-16 | Last 8 layers | | | 17,256,715 | 99 | 98 |
| Model 5 | VGG-16 | Last 12 layers | 128 | 128 | 18,732,043 | 97 | 89 |
| Model 6 | VGG-19 | None | | | 4,277,515 | 98 | 92 |
| Model 7 | VGG-19 | All | | | 24,301,899 | 97 | 88 |
| Model 8 | VGG-19 | Last 4 layers | | | 11,356,939 | 99 | 92 |
| **Model 9** | **VGG-19** | **Last 8 layers** | | | **18,436,363** | **100** | **99** |
| Model 10 | VGG-19 | Last 12 layers | | | 22,566,411 | 97 | 88 |

However, from this table, it can be noticed that the proposed model, i.e., Model 9 with retraining last eight layers of VGG19 integrated with LSTM and fully connected layer, achieved the maximum F1 score of 99%, which is the ultimate evaluation metric considered in this paper. In this study, dropout rate of 0.2 is applied to the fully connected layer to reduce overfitting of the model. The rate 0.2 means 20 neurons out of each 100 from this layer are ignored while training. Moreover, our proposed model contains 18,436,363 weights/parameters, whereas Model 7 contains 24,301,899 weights/parameters, in which we fine-tuned all layers of VGG-19. Here, it can be observed that our proposed model contains 5,865,536 fewer weights/parameters than Model 7. The feature maps produced by the convolutional part of Model 9 from a frame of videos of each class are depicted in Figure 8.

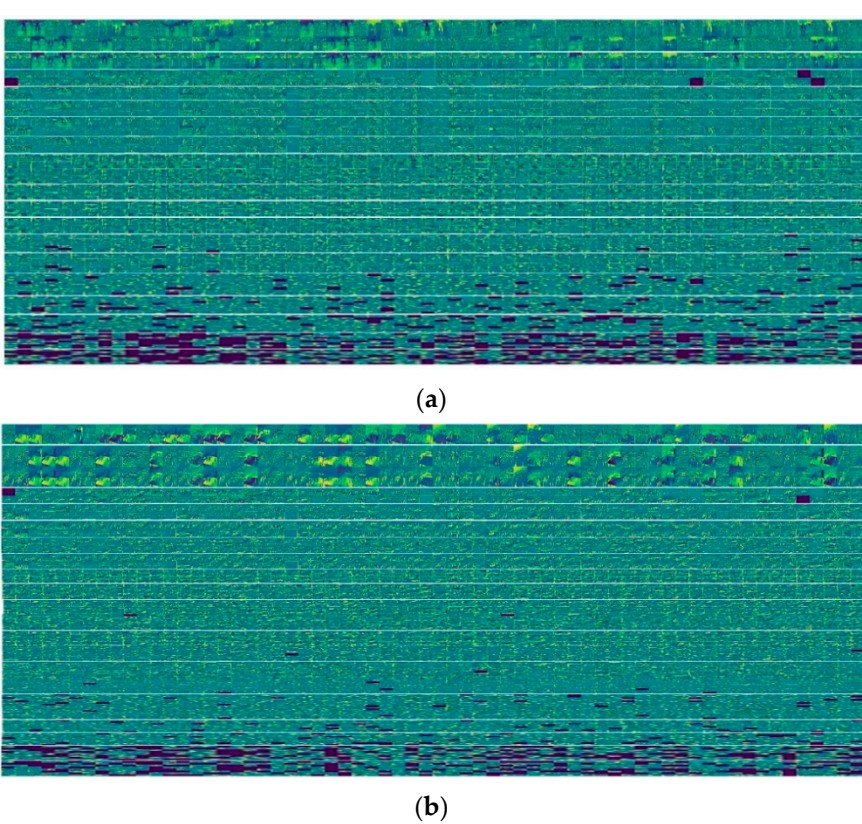

(**a**)

(**b**)

**Figure 8.** *Cont.*

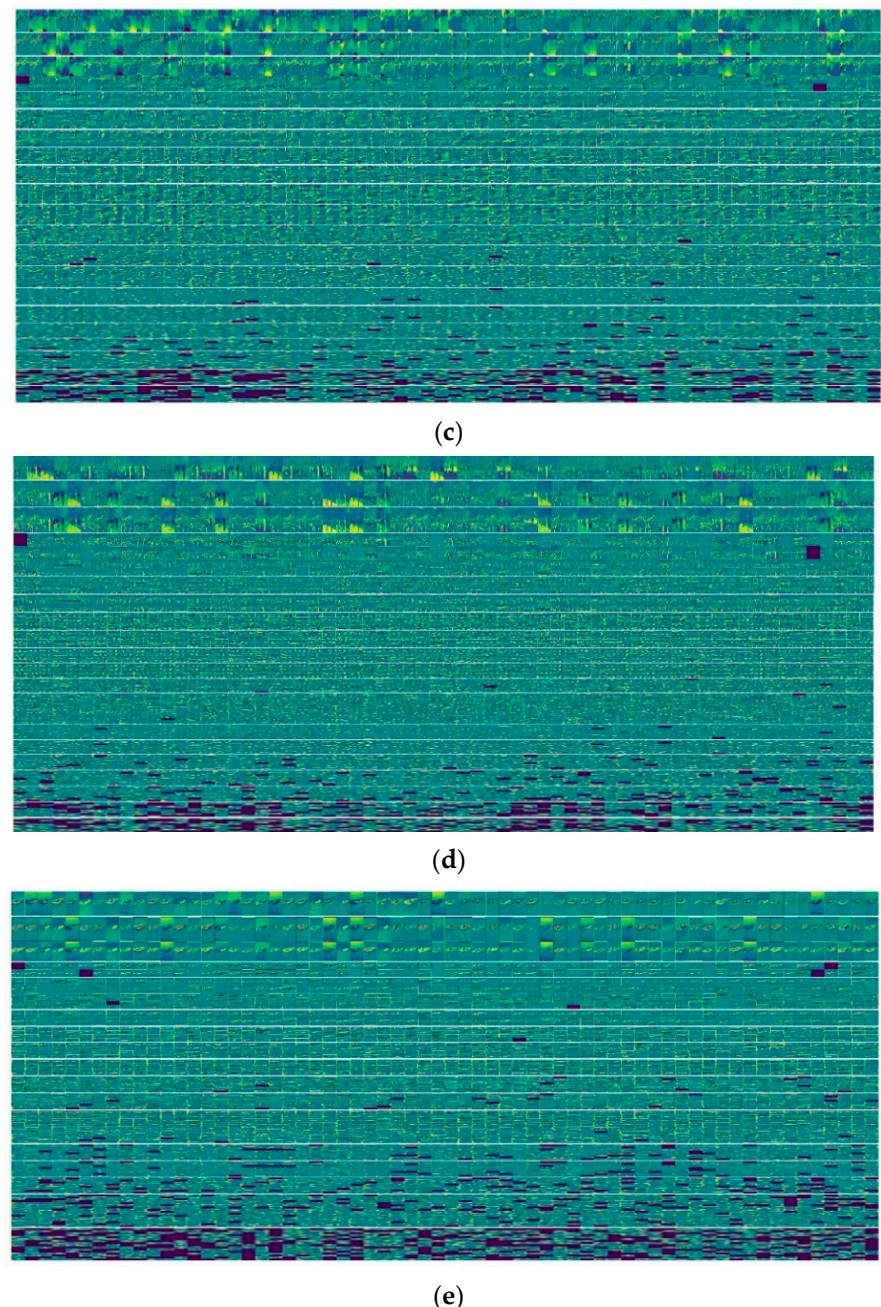

**Figure 8.** The feature maps produced by the convolutional part of Model 9 of (**a**) Boli Khela, (**b**) Kabaddi, (**c**) Kho Kho, (**d**) Lathi Khela, and (**e**) Nouka Baich.

The feature map provides a glimpse of how the input data is dissected into various filters of the model. However, deep neural networks work like a black-box in extracting features from the input. Higher-level layers produce feature maps that are more abstract and opaquer to humans than lower-level layers in the model. They encode higher-level features that are difficult to extract and present in a human-readable way [29]. Therefore, the deep learning models extract higher-level features that are mystical and more immense in amount than the features considered by mere humans.

There are certain key distinguishable features in each sports class. Boli Khela is mostly played between two competitors in a sandy wrestling playground, whereas the Kabaddi playground is divided into two halves occupied by two opposing teams. However, the chasing team in Kho Kho assembles in eight squares in the rectangle playground's central lane. Additionally, the Lathi Khela is a stick fighting sport played between rival groups.

Moreover, the Nouka Baich is a traditional boat rowing sport mostly played during the rainy season on water surfaces. In this paper, the feature maps for each class represented in Figure 8 are organized according to feature maps of the lower-level to higher-level layers of the convolutional part of Model 9. From the feature maps of lower-level layers, it can be observed that our convolutional architecture primarily grabs the surrounding of the playground, key sports equipment, and characteristics of players of each sport as features. As the layers go deeper, the extracted features become more encoded and lack human readability.

The training and validation accuracy, along with the loss curves of the proposed model, is represented in Figure 9. It can be noticed from this figure that the maximum training accuracy of the model was attained at the 27th epoch, where validation accuracy was the maximum, i.e., 99%.

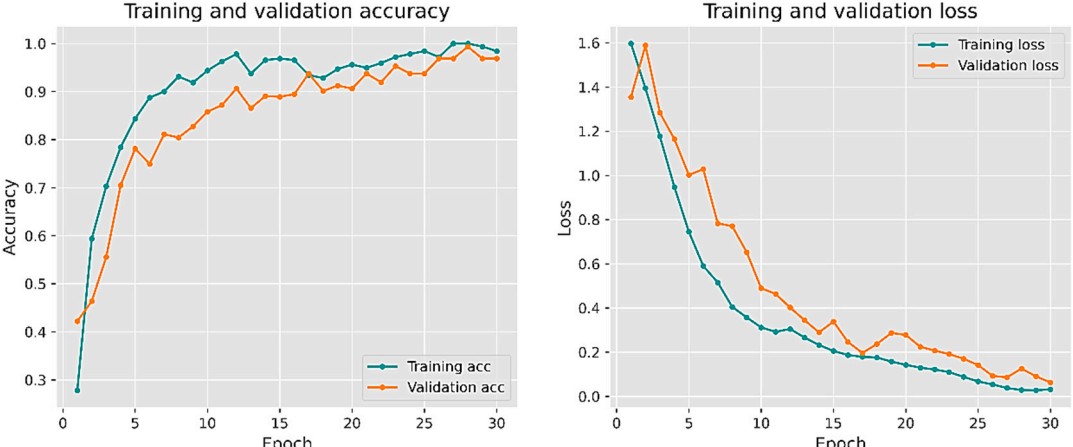

**Figure 9.** Performance of Model 9.

The confusion matrix of this proposed model, i.e., Model 9, is portrayed in Figure 10. The figure shows that, due to having some similar semantic characteristics, only a test sample in "Kabaddi" class was misclassified as "Boli Khela", which proves this model's efficiency to a great extent. To get more insight into Model 9's performance, class-wise values of evaluation metrics are demystified in Table 6.

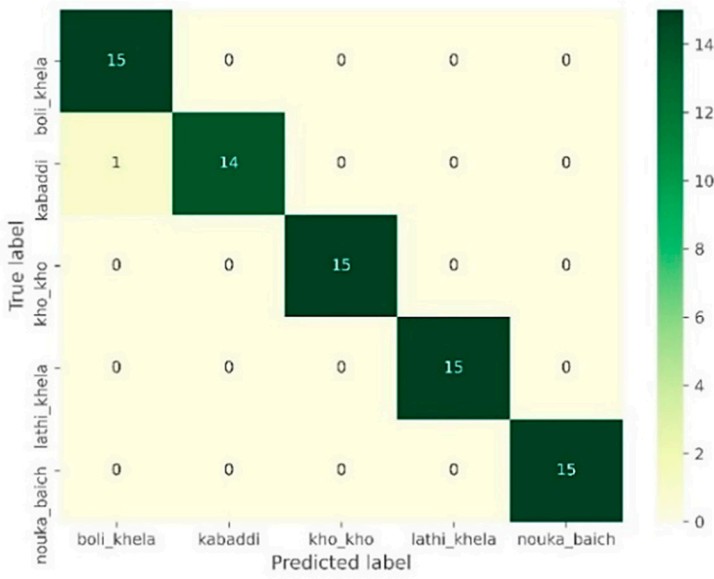

**Figure 10.** Confusion matrix for Model 9.

**Table 6.** Class-wise performance of Model 9.

| Classes | Mean Accuracy (%) | Precision (%) | Sensitivity (%) | Specificity (%) | F1 Score (%) |
|---|---|---|---|---|---|
| Boli Khela | | 94 | 100 | 98 | 97 |
| Kabaddi | | 100 | 93 | 100 | 97 |
| Kho Kho | 99 | 100 | 100 | 100 | 100 |
| Lathi Khela | | 100 | 100 | 100 | 100 |
| Nouka Baich | | 100 | 100 | 100 | 100 |

Driven by the impressive performance of Model 9 over the TBSV dataset, we extended our research work by employing Model 9 in the classification task on the four most prominent datasets: KTH, UCF-11, UCF-101, and UCF Sports. For all the datasets, the split ratio is considered as 80:20 (i.e., 80% of the video data is selected for training, and 20% for testing the network randomly). For consistency, 10 frames per video are considered in the experiment. Table 7 represents the performance comparison of our proposed model over KTH, UCF-11, and UCF Sports datasets with some recent works. On the KTH dataset, our model achieved better accuracy, i.e., 97%, than the approach of Jaouedi et al. [18], which was based on a hybrid deep learning. In the case of the UCF-11 dataset, the proposed model achieved 95.6% accuracy, which surpasses the performance of the method of Ge et al. [19], which was based on attention mechanism-based CNN-LSTM network. Additionally, after training the proposed model over the UCF Sports dataset, 94% test accuracy has been acquired, proving this model's worthiness relative to the DCNN-LSTM based method of Cheng et al. [30] and deep convolution network-based PCANet with encoding method of Abdelbaky and Aly [20].

**Table 7.** Performance comparison with other methods.

| KTH | | UCF-11 | | UCF Sports | |
|---|---|---|---|---|---|
| **Method** | **Accuracy** | **Method** | **Accuracy** | **Method** | **Accuracy** |
| Latah [31] | 90.34% | Meng et al. [32] | 89.7% | Zare et al. [33] | 82.4% |
| Abdelbaky and Aly [20] | 93.33% | Yang et al. [21] | 91.2% | Jaouedi et al. [18] | 89.01% |
| Xu et al. [34] | 95.80% | Ullah et al. [16] | 92.84% | Abdelbaky and Aly [20] | 90% |
| Jaouedi et al. [18] | 96.30% | Ge et al. [19] | 94.12% | Cheng et al. [30] | 90% |
| Our Proposed Model | 97% | Our ProposedModel | 95.6% | Our Proposed Model | 94% |

Moreover, our proposed model achieved 96% accuracy on UCF-101 dataset and outperforms [18], which also recognizes 66 classes of this dataset by hybrid deep learning model and achieved 89.30% accuracy. Moreover, in [35], 81.93% accuracy was obtained in recognizing 101 classes of this dataset using deep multimodal fusion features. Relative to this, because of our limitations of hardware and memory configuration, we have considered only 66 classes of two categories (sports and body-motion only) in UCF-101 dataset and achieved 96% accuracy.

## 5. Conclusions

In this paper, a combined network consisting of CNN and RNN incorporated with a multiclass classifier is proposed to classify traditional Bangladeshi sports videos. This task is quite challenging, as it requires the capture and processing of both spatial and temporal features. However, the main barrier to this task is the scarcity of the Traditional Bangladeshi Sports Video (TBSV) dataset. Hence, we have built a new traditional Bangladeshi sports video dataset containing 500 sports videos of five categories: Boli Khela, Kabaddi, Lathi Khela, Kho Kho, and Nouka Baich, which is one of the key contributions of this work. Indeed, our research can serve a noble cause of reviving the lost traditional Bangladeshi sports and thus spreading their cultural significance. To accomplish the classification task on the TBSV dataset, a scratch network is developed by incorporating CNN with LSTM,

which showed decent accuracy of 91% over the test data. On the other hand, an experiment was conducted with the transfer learning approach through fine-tuning VGG16 and VGG19 networks. However, the proposed model is comprised of the fine-tuned VGG19 network integrated with LSTM, which shows promising achievement on the TBSV dataset and some benchmark datasets of human activity: KTH, UCF-11, UCF-101, and UCF Sports. Additionally, the experimental results testify that our proposed model can be used in sports and human activity recognition tasks to a great extent. However, there is still space for improvement in this field. In the future, we aim to work with some more traditional Bangladeshi sports classes to enhance the extensiveness of this work. Furthermore, for better generalization, some more augmentation techniques are intended to use. Some ensemble models are aimed to implement in future, as in [22], with in-built regularization technique. Additionally, we aspire to explore the fine tuning of some other pretrained models like Resnet, Inception, Xception, etc., and analyze their potency in this ground.

**Author Contributions:** Conceptualization: K.D.; data curation: M.S.S.; formal analysis: M.S.S.; investigation: M.S.S.; methodology: M.S.S.; software: M.S.S.; supervision: K.D.; validation: M.S.S.; visualization: M.S.S.; writing—original draft: M.S.S.; writing—review and editing: K.D., P.K.D. and T.K. All authors have read and agreed to the published version of the manuscript.

**Funding:** This research received no external funding.

**Institutional Review Board Statement:** Not applicable.

**Informed Consent Statement:** Not applicable.

**Data Availability Statement:** The authors have constructed a novel dataset named Traditional Bangladeshi Sports Video (TBSV) dataset for the experiment, which is available on request from the corresponding author. Moreover, the authors have used publicly archived KTH, UCF-11, UCF Sports, and UCF-101 datasets for validating the experiment. The KTH dataset is available in [24]. The UCF-11 dataset is available in [25]. The UCF Sports dataset is available in [27,28]. The UCF-101 dataset is available in [26].

**Conflicts of Interest:** The authors declare no conflict of interest.

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
