# Peer review of "Traditional Bangladeshi Sports Video Classification Using Deep Learning Method"

_applsci, doi:10.3390/app11052149_

Round 1
Reviewer 1 Report
- The whole paper is well organized and describes the details of training in practice in great detail. The preprocessing is important for the training of machine learning, and the results are listed on the Table.
- The selection of CNN backbone is based on a well consideration. First is to select the backbone network and then set the target of fine tune options. Then it is to find out the best model, and run on other datasets. Finally it is to get the good result. In overall the procedure is fine, but it looks the novelty on this work is somehow limited.
- The visualization of feature maps is a well-organized comparison figure. You can see that the feature map is more readable at lower levels. Conversely, the higher the level, the less understandable it is.
- The Table 7., it doesn’t compare enough data. It is suggested to compare with at least 2 more papers related to the use of CNN or LSTM methods in action recognition on these dataset.
- About Table 7. of UCF-101, you have mentioned before that you only classify 2 categories totaling 66 classes, and only use 66 class classification accuracy to compare with others. Maybe the text should be explained in more detail.
Reviewer 2 Report
This work presents an interesting exercize of applying a complex CNN+LSTM model for video predicition. Some clarification comments:
- It is not clear how the architecture has been implemented in tensorflow. Using a TimeDistributed layer?
- How many weigths (parameters) has the model?
- It is not clear if the CNN stack shares their weigths to avoid overfitting. Please, clarify.
- Considering the problem of overfitting some ensemble models could be interesting. Have they been considered? Or, as reference,related works. For example: IoT type-of-traffic forecasting method based on gradient boosting neural networks.
